# The Role of Merlin/NF2 Loss in Meningioma Biology

**DOI:** 10.3390/cancers11111633

**Published:** 2019-10-24

**Authors:** Sungho Lee, Patrick J. Karas, Caroline C. Hadley, James C. Bayley V, A. Basit Khan, Ali Jalali, Alex D. Sweeney, Tiemo J. Klisch, Akash J. Patel

**Affiliations:** 1Department of Neurosurgery, Baylor College of Medicine, Houston, TX 77030, USA; sungho.lee@bcm.edu (S.L.); patrick.karas@bcm.edu (P.J.K.); caroline.hadley@bcm.edu (C.C.H.); james.bayley@bcm.edu (J.C.B.V.); abdul.khan@bcm.edu (A.B.K.); ali.jalali@bcm.edu (A.J.); 2Department of Otolaryngology-Head and Neck Surgery, Baylor College of Medicine, Houston, TX 77030, USA; alex.sweeney@bcm.edu; 3Jan and Dan Duncan Neurological Research Institute, Texas Children’s Hospital, Houston, TX 77030, USA; tiemo.klisch@bcm.edu; 4Department of Molecular and Human Genetics, Baylor College of Medicine, Houston, TX 77030, USA

**Keywords:** meningioma, merlin, neurofibromin 2, next-generation sequencing, DREAM complex

## Abstract

Mutations in the *neurofibromin 2* (*NF2*) gene were among the first genetic alterations implicated in meningioma tumorigenesis, based on analysis of neurofibromatosis type 2 (NF2) patients who not only develop vestibular schwannomas but later have a high incidence of meningiomas. The *NF2* gene product, merlin, is a tumor suppressor that is thought to link the actin cytoskeleton with plasma membrane proteins and mediate contact-dependent inhibition of proliferation. However, the early recognition of the crucial role of *NF2* mutations in the pathogenesis of the majority of meningiomas has not yet translated into useful clinical insights, due to the complexity of merlin’s many interacting partners and signaling pathways. Next-generation sequencing studies and increasingly sophisticated *NF2*-deletion-based *in vitro* and *in vivo* models have helped elucidate the consequences of merlin loss in meningioma pathogenesis. In this review, we seek to summarize recent findings and provide future directions toward potential therapeutics for this tumor.

## 1. Introduction

Meningiomas are the most common type of primary intracranial tumors, currently classified by the World Health Organization (WHO) as benign (grade I), atypical (grade II), or malignant (grade III). Over the years, our understanding of the molecular underpinnings of these tumors has been greatly accelerated by advancements in next-generation sequencing (NGS) technology. It has been thought for some time that the protein encoded by the neurofibromin 2 (*NF2*) gene is a key to understand these brain tumors. The *NF2* gene resides on the long arm of chromosome 22 (chr22q) and encodes a 69 kDa protein called merlin (moesin-ezrin-radixin-like protein), which is a part of the Band 4.1 FERM gene family [1]. Merlin is a cytoskeleton scaffolding protein that links actin filaments, transmembrane receptors, and intracellular signaling molecules to regulate several essential pathways controlling proliferation and survival. These include the hippo pathway, mammalian target of rapamycin (mTOR)/PI3K/AKT pathway, and receptor tyrosine kinases (RTKs) [2,3]. In this review, we summarize the role of *NF2* loss in meningioma pathogenesis and its impact on meningioma biology based on the known functions of merlin. 

## 2. Evidence Linking *NF2* and Meningiomas—Inherited Disorders 

### 2.1. Neurofibromatosis Type 2 (NF2)

The first indication that meningiomas may have a genetic contribution came from neurofibromatosis type 2 (NF2) [4,5]. NF2 is a rare autosomal dominant tumor syndrome with an estimated birth incidence of 1 in 33,000 [6], resulting from biallelic inactivation of *NF2*. Development of bilateral vestibular schwannomas is a pathognomonic feature present in approximately 60% of cases [7]. However, considerable heterogeneity in clinical presentation has led to the development of additional diagnostic criteria [8] (Table 1). Other lesions encountered in NF2 include non-vestibular schwannomas, meningiomas, ependymomas, and congenital cataracts.

The *NF2* mutational spectrum in NF2 is vast; most variations do not recur. Nonsense (39%) and frameshift (27%) mutations in the *NF2* gene are most frequent in NF2 patients, with splice site (25%) and non-truncating mutations (7%) making up a smaller fraction of mutations [9]. Nonsense and frameshift mutations that truncate the protein are associated with a more severe disease phenotype, including increased frequency of multiple and recurring meningiomas [10,11,12,13,14]. Somatic mosaicism occurs in up to 33% of NF2 patients, which results in a milder phenotype and lower risk of transmission to offspring [15,16,17]. Mutations may only be detectable in the tumor tissue, emphasizing the importance of analyzing surgical specimens.

Intracranial meningiomas affect about half of NF2 patients [8,10,18,19], and spinal meningiomas are seen in approximately 20% of the patients [20]. Over half of these patients have multiple meningiomas that exhibit heterogeneous behavior and an asynchronous growth pattern [21]. While the majority of syndromic meningiomas remain stable in size or grow minimally over time, few tumors, including tumors that appear *de novo*, grow more rapidly and therefore are more frequently resected [22]. Furthermore, several distinct histological subtypes were seen in patients who underwent resection of multiple meningiomas, suggesting that *NF2* inactivation is an early tumorigenic event that occurs prior to commitment to a specific histopathologic subtype. Subsequent studies in sporadic meningiomas demonstrated that up to 60% of these cases exhibit inactivation of *NF2* by somatic mutation, epigenetic inactivation, or allelic loss of chr22q [23,24,25,26]. These findings suggest that *NF2* loss is a critical event in the development of a subpopulation of meningiomas. 

### 2.2. Schwannomatosis

Schwannomatosis is characterized by the development of multiple schwannomas in the absence of other NF2-defining lesions such as bilateral vestibular schwannomas or ependymomas [27]. It is a rarer disorder than NF2, with an estimated incidence of 1/40,000 to 1/70,000 [28]. Germline mutations in SWItch/Sucrose Non-Fermentable (SWI/SNF)-related matrix-associated actin-dependent regulator of chromatin subfamily B member 1 (*SMARCB1*) or leucine-zipper-like transcriptional regulator 1 (*LZTR1*) predispose to the disorder [29,30]. Both genes are located on chr22q in proximity to *NF2*. In the presence of *SMARCB1* or *LZTR1* mutations, there is also biallelic *NF2* loss of function, via acquired somatic mutations and loss of heterozygosity (LOH) [31]. Multiple meningiomas occur in 5% of schwannomatosis cases [32], but only in association with *SMARCB1* mutations [33,34,35], hinting at a potential interaction between NF2 and SMARCB1 in meningioma pathogenesis, later validated in NGS studies.

## 3. Insights from NGS Studies

Over the past decade, several groups have leveraged the wide availability and reduced cost of NGS to characterize meningiomas, providing additional insights into the mutation landscape of meningiomas from early genetic studies (Table 2). 

Several whole-exome sequencing (WES) studies identified recurrent somatic mutations in *NF2*, TNF receptor-associated factor 7 (*TRAF7*), Krupple-like factor 4 (*KLF4*), v-akt murine thymoma viral oncogene homolog 1 (*AKT1*), smoothened, frizzled family receptor (*SMO*), and RNA polymerase II subunit A (*POLR2A*) in benign (WHO grade I) tumors [36,37,39]. Interestingly, a significant proportion (6.4%) of *NF2*-mutated tumors harbored recurrent mutations in *SMARCB1* [39]. While *KLF4* or *AKT1* mutations almost always co-occurred with *TRAF7* mutations, they did not occur together; *SMO*-mutated meningiomas all harbored the activating L412F mutation [36]. All secretory meningiomas carried *TRAF7*/*KLF4* mutations [36,38]. One study of 18 radiation-induced meningiomas found that nearly all had chr22q LOH, with a majority also having *NF2* loss via fusion events [40]. All of these studies have found that chr22q/*NF2* loss does not co-occur with *TRAF7*, *AKT1*, *KLF4*, *SMO*, or *POLR2A* mutations, which were all found in benign meningiomas (Figure 1) [36,37,38,39,40].

On the other hand, a different mutational landscape is found in high-grade (WHO grade II and III) meningiomas, which demonstrate higher recurrence and shorter survival compared to benign meningiomas [26,44,45,46]. Two recent NGS studies of high-grade meningiomas found more chromosomal abnormalities compared to benign meningiomas, similar to other aggressive cancers [41,42]. Most high-grade meningiomas were characterized by *NF2* loss, without any other significantly recurring somatic mutations, in contrast to benign meningiomas [41,42]. 

Bi *et al.* examined 36 paired samples of meningiomas undergoing malignant progression and found *NF2* loss in 73% of the cases in both low- and high-grade samples from the same patient, emphasizing that *NF2* loss is an early event in meningioma progression. Losses of chromosome arms 1p, 6q, 14q, 3p, 10q, 18q, and 19q were additionally seen in these tumors. Interestingly, recurrent tumors from the same patient demonstrated 75% overlap of arm-level somatic copy number variations (CNV), suggesting that these chromosomal losses are again an early and essential feature in high-grade meningiomas.

Harmanci *et al.* found that the majority of atypical meningiomas had *NF2* loss and fell into two categories: those with significant chromosomal losses (“CNV-high”) and those with *SMARCB1* mutations (“CNV-low”). However, they did not find any differences in the transcriptional profiles of these two groups, finding all of these tumors significantly enriched for cell cycle processes, including upregulation of E2F and FOXM1 networks. Additionally, they found *NF2* mutant tumors demonstrated increased methylation of polycomb repressive complex 2 (PRC2) target genes, driven by the upregulation of EZH2, the catalytic subunit. Vasudevan *et al.* sought targetable pathways in high-grade meningiomas and found that elevated *FOXM1* expression is associated with poor clinical outcomes [47].

Recently, we undertook a large-scale RNA-seq based study of 160 tumors of all grades to further refine our understanding of meningioma biology [43] (Figure 1). Using only transcriptional profiling, we found three types of meningioma. Subsequent analysis of their genomic landscape revealed one type (type A) was composed of only benign meningiomas with mutations in *TRAF7*, *AKT1*, *KLF4*, *SMO*. The other two types had chr22q/*NF2* loss (type B and C) but no other known mutations. Interestingly, these two types had very different molecular and clinical characteristics, arguing that merlin inactivation sets the stage for additional tumorigenic events that have dramatically different consequences. While type A and B tumors did not recur after complete resection, type C tumors (a mixture of WHO grade I, II, and III tumors) behaved “aggressively”, recurring frequently despite complete resection. An important outcome of this was that the RNA-seq based classification system could identify WHO grade I meningiomas that behaved aggressively despite their benign histopathologic features.

Type B (a mixture of grade I and II tumors) meningiomas were characterized by chr22q/*NF2* loss and co-occurrence of *SMARCB1* mutations [43]. The atypical meningiomas that Harmanci et al. referred to as “CNV-low” likely represent type B tumors [42]. Gene set enrichment analysis (GSEA) suggests that type B tumors are characterized by loss or dysfunction of the repressive PRC2 complex, which is responsible for H3K27 di- and trimethylation and subsequent chromatin silencing. In type B tumors, co-immunoprecipitation of EZH1, a core subunit of the PRC2 complex, did not pull down the other critical PRC2 subunits such as EED and SUZ12, suggesting a failed assembly of the PRC2 complex. In addition, a portion of type B tumors harbored *SMARCB1* mutations. SMARCB1, whose gene is also encoded on chr22q, is a critical component of the SWI/SNF complex. The SWI/SNF and PRC2 chromatin remodeling complexes have a close and complex interconnectivity in regulating the chromatin state [48,49]. How these complexes are dysfunctional in type B meningiomas remains unknown, which highlights the need for further studies to understand the complex biology underlying these tumors. It will be interesting to understand how the di- and trimethylation profiles in the three types are altered.

On the other hand, type C meningiomas, also with *NF2* loss, had a significant burden of chromosomal gains/losses, most commonly loss of chr22q and chr1p together, as well as significantly shorter recurrence-free survival, despite over half of these tumors being WHO grade I [43]. The “CNV-high” atypical meningiomas reported by Harmanci *et al.* likely correspond to these tumors [42]. GSEA showed that the target genes of the DREAM complex, including *FOXM1* and *MYBL2*, were significantly enriched in type C tumors, when compared to the other two types. The DREAM complex is a highly conserved master regulator of the cell cycle which can alternate between a repressive form that inhibits cell cycle gene expression and an activated form that promotes the progression of the cell cycle, on the basis of the interaction of the MuvB core with RB-like proteins (e.g., E2F2) or FOXM1 with MYBL2, respectively [50]. These findings provide further clarity to the observation that increase in the cell cycle-focused E2F2 transcriptional network and elevated expression of FOXM1 are associated with high-grade meningiomas [42,47]. Interestingly, while *RBBP4*, one of the MuvB core complex members, is located on chr1p, its expression was not significantly changed in type C tumors. How chr1p loss might lead to the switch from the repressor form of the DREAM complex to the activator form remains to be discovered.

Taken together with previous work, these results suggest that chr22q/*NF2* loss is a requisite for the development of aggressive meningiomas which also harbor chr1p losses. *NF2* mutation and chromosomal losses may be two distinct processes that work in parallel, but they are early events in tumorigenesis. This also explains the findings highlighted by Dewan *et al.* in which two tumors within a single NF2 patient can have dramatically different clinical courses [51].

## 4. Merlin Signaling

### 4.1. Molecular Conformation

Early insights into merlin function were gleaned from its sequence homology to the FERM family of proteins. Canonical ERM proteins are comprised of a N-terminal FERM domain, followed by an alpha-helix domain and a C-terminal domain. Typically, ERM proteins are maintained in a dormant state by an intramolecular association between N-terminal FERM and C-terminal domains. Upon phosphorylation of a conserved threonine residue at the C-terminal domain by Rho kinase, ERM proteins undergo a conformation change, unmasking sites at the C-terminal domain for the binding of F-actin and other membrane proteins, thereby rendering it active [52,53].

While bearing some similarity to other ERM proteins, the process by which merlin undergoes conformational change and the relative importance of its open and closed state to its scaffolding and tumor suppressor function remain a source of controversy. Traditionally, phosphorylation of merlin at Serine 518 by p21 activated kinase 1 (PAK1) or protein kinase A (PKA) versus dephosphorylation mediated by MYPT1–PP1 was believed to mediate the transition between its open and closed conformations [54,55,56]. Initial studies demonstrated that the tumor suppressor activity of merlin was dependent upon the non-phosphorylated, closed conformation of merlin. However, detailed structural analyses of merlin using fluorescence energy transfer analysis showed that the hyperphosphorylation of Serine 518 or the expression of phosphomimetic and non-phosphorylatable S518D and S518A mutations had only subtle effects on the conformation of merlin [57]. Moreover, studies based on small-angle neutron scattering and immunoprecipitation showed that the phosphorylation of Serine 518 stabilized the closed form rather than promoting an open conformation as initially thought, and that the phosphorylated form of merlin was nonetheless able to interact with its target proteins [58]. Two recent studies reported the interaction between merlin and phosphatidylinositol 4,5-bisphosphate (PIP_2_), which promotes its open conformation and anti-proliferative activity, whose effects may be distinct from those mediated by phosphorylation at Serine 518 [57,59]. Clearly, the mechanism of the conformational change of merlin is different from that of typical FERM proteins. Merlin’s activity might depend on other factors and, hence, remains an interesting and active area of investigation.

### 4.2. Contact Inhibition

Merlin functions as a tumor suppressor in a wide range of cancers. However, relatively few studies have investigated its molecular mechanisms specific to meningioma pathogenesis. Consequently, the understanding of merlin signaling in meningiomas needs to be supplemented by insights from studies in other cancers and cell types.

Early studies suggested that merlin’s tumor suppressor activity is related to its contact inhibition of proliferation. In various cell types, merlin is upregulated and hypophosphorylated with an increasing degree of cell confluency [54]. Dephosphorylated, active merlin preferentially interacts with and inhibits CD44, a cell-to-cell adhesion molecule and receptor for hyaluronan, an abundant extracellular matrix (ECM) component [60]. Treatment of sub-confluent schwannoma cells with hyaluronan rapidly induced dephosphorylation of merlin and inhibited cell growth, which was abolished by mutation in the hyaluronan binding domain of CD44. CD44 also constitutively associates with various RTKs that mediate growth factor signaling. Consistent with this notion, siRNA-mediated knockdown of merlin in schwannoma cells increased the levels of ErbB2/ErbB3 RTK, suggesting that merlin normally functions to reduce the availability of RTKs at the plasma membrane [61].

In addition, merlin-mediated contact inhibition is critically regulated by a reciprocal interaction with Ras-related C3 botulinum toxin substrate 1 (RAC1) and its downstream kinase PAK1 (Figure 2). In confluent cells, activation of PAK1 by RAC1 is sufficient to release the cells from contact inhibition [55]. However, active (dephosphorylated) merlin suppresses the recruitment of RAC1 to the plasma membrane, preventing the activation of RAC1 and PAK1. On the other hand, PAK1 phosphorylates merlin at Serine 518, and its subsequent inactivation prevents the translocation to the plasma membrane, mitigating the inhibitory effect of merlin on RAC1. Merlin-deficient meningioma cell lines demonstrate increased expression of PAK1 compared to normal arachnoid cap cells, and knocking down PAK1 expression using doxycycline-inducible shRNA or treatment with PAK1 inhibitors inhibited cell proliferation in vitro and tumor growth in xenograft models [62]. 

### 4.3. Hippo Pathway

The regulation of the Hippo pathway by merlin is better characterized in meningiomas. This evolutionarily conserved signaling pathway, first identified in *Drosophila melanogaster*, inhibits cell proliferation and promotes apoptosis to limit organ size during normal development and suppress tumorigenesis [63]. It relies upon a kinase cascade including macrophage-stimulating 1/2 (MST1/2), salvador family WW domain-containing protein 1 (SAV1, also called WW45), and large tumor suppressor 1/2 (LATS1/2) to phosphorylate yes-associated protein (YAP), leading to the sequestration of this key transcriptional coactivator from the nucleus, thereby inhibiting the transcription of target genes associated with proliferation and survival (Figure 2).

The suppression of merlin using NF2 siRNA in established meningioma cell lines inhibited contact-dependent inhibition of growth and promoted cell cycle progression in association with increased levels and elevated nuclear co-localization of YAP [64]. In addition, nuclear YAP immunoreactivity was revealed in 92% of merlin-negative tumors, further suggesting that merlin is a negative regulator of the Hippo pathway in meningiomas. A separate study also confirmed a complementary pattern of merlin expression and nuclear YAP expression, although, unexpectedly, nuclear YAP expression was found even in merlin-positive tumors [65]. Furthermore, nuclear YAP expression was increased when meningioma cell lines were plated at sparse cell density and in less rigid extracellular matrix, suggesting that inhibition of YAP-mediated Hippo signaling pathway by merlin is dependent on cell-to-cell contact and upstream adhesion molecules. 

### 4.4. PI3K/AKT/mTOR Pathway

Phosphoinositide 3-kinase (PI3K)/AKT signaling is involved in the regulation of cell growth and proliferation [66]. Growth factor stimulation triggers the production of phosphatidylinositol (3,4,5) triphosphate (PIP3) by PI3K, leading to the phosphorylation and activation of downstream AKT at the plasma membrane and subsequent activation of mammalian target of rapamycin complex (mTORC), resulting in the translation of target proteins. It has been shown that merlin inhibits the activation of PI3K by binding phosphatidylinositol 3-kinase enhancer-L (PIKE-L) [67]. The finding that a subset of meningiomas have activating E17K mutations in AKT supports a crucial role of this pathway in meningioma biology [36,37,39,43]. High-grade meningiomas showed higher levels of phosphorylated AKT compared to benign tumors, further supporting a role for this pathway in merlin-driven meningioma pathogenesis [68]. Conversely, inhibition of AKT phosphorylation decreased meningioma growth in several in vitro studies [69,70].

Merlin has also been identified as a negative regulator of mTORC1 (Figure 2). This association was initially suspected from the observation that cultured primary human merlin-deficient meningioma cells exhibited a strikingly enlarged morphology compared to non-neoplastic arachnoid cap cells from the same patient [71], bearing similarity to tuberous sclerosis, wherein mutations in TSC1 and TSC2 lead to aberrant activation of mTORC1 [72]. Additional studies demonstrated that merlin-deficient primary meningioma cell lines and tumors exhibit constitutive activation of mTORC1, and conversely, exogenous expression of wild-type, but not mutant, merlin inhibited mTORC1 signaling [73]. Interestingly, merlin-deficient mTORC1 activation was independent of upstream PI3K/Akt and ERK signaling, which traditionally activate this signaling in response to various mitogenic stimuli. Therefore, the non-canonical mechanism by which the loss of merlin induces mTORC1 signaling is unknown.

mTORC1 inhibition is a validated therapeutic strategy in various types of cancers, and several orally bioavailable mTORC1 inhibitors are currently FDA-approved, including temisirolimus and everolimus. Despite the incomplete understanding of the interaction between merlin and mTORC1, mTORC1 inhibitors have been tested in various in vitro and in vivo meningioma models and patients. Temisirolimus and everolimus treatment significantly decreased viability and proliferation of a meningioma cell line in a concentration-dependent manner, and temisirolimus significantly reduced tumor burden in xenograft models [74]. Interestingly, shRNA-mediated downregulation of merlin rendered the meningioma cells more resistant to mTORC1 inhibition, presumably due to merlin deficiency-mediated constitutive upregulation of mTORC1 activity. In a small prospective phase 2 trial of 17 patients with progressive or refractory symptomatic meningiomas, concurrent treatment with bevacizumab and everolimus demonstrated overall median progression-free survival of 22 months [75]. An additional trial with the mTORC1/2 inhibitor AZD2014 is currently ongoing for patients with neurofibromatosis type 2-associated meningiomas (NCT 02831257) and recurrent high-grade meningiomas (NCT 03071874).

## 5. Animal Models 

### 5.1. Xenograft Models

Xenograft models rely upon the implantation of human meningioma cells into immunocompromised mice. Their usefulness as a tool to investigate the role of merlin in meningioma pathogenesis is limited by several factors. Primary cells isolated from surgical samples, especially from benign tumors, do not reliably generate tumors, and the methods for their intracranial delivery have not been standardized [76]. In our experience, intracranial implantation of high-grade meningiomas can lead to xenograft formation; however, these do not serially transplant. Thus, large amounts of the original tumor are necessary to continue to develop numerous xenografts from a single tumor.

Most investigators have therefore relied upon established cell lines for xenograft experiments, such as the well-characterized BenMen1 line derived from a WHO grade I meningothelial meningioma which recapitulates key histologic and genetic features of the parent tumor, including *NF2* mutation [60,61]. However, this cell line has been retrovirally transduced with the human telomerase reverse transcriptase (*hTERT*) gene, in order to bypass senescence, with unclear alterations in underlying tumor biology [77]. The less well characterized HBL-52 cell line was derived from a transitional grade I optic canal meningioma, but this cell line harbors the *TRAF7* driver mutation [78,79,80]. Other cell lines are derived from high-grade meningiomas such as the IOMM-Lee and CH-157MN cell lines, and these cell lines demonstrate the genomic instability seen in more aggressive parent tumors [61,64,65]. However, the IOMM-Lee cell line has intact *NF2*, rendering it unsuitable for studies of merlin function [79]. Overall, these cell lines have no comparable controls, and their predetermined or laboratory-altered genetics are unlikely to account for the full complexity of their real-life counterparts. Finally, due to the immunocompromised nature of the host, potentially important immune interactions cannot be analyzed.

### 5.2. Genetically Engineered Mouse Models (GEMM)

Homozygous deletion of *Nf2* in mice is embryonically lethal, and heterozygous *Nf2* knockout mice develop osteosarcomas but not meningiomas [81]. Therefore, initial efforts to create a *Nf2* deletion-based GEMM of meningiomas relied upon a conditional knockout approach. Given the lack of known arachnoid-specific promoters at the time, an adenovirus encoding recombinant Cre was injected into the subdural space of mice harboring two copies of floxed *Nf2* allele (*Nf2*^flox/flox^), driving arachnoid-specific deletion of *Nf2*. Remarkably, this was sufficient to induce a range of benign meningioma encompassing the transitional, meningothelial, and fibroblastic subtypes, although only a minority of injected mice ultimately developed meningiomas [82]. Additional studies demonstrated that the arachnoid-specific deletion of cyclin-dependent kinase inhibitor (*Cdkn2ab*), frequently deleted in high-grade meningiomas, increased meningioma frequency and the development of grade II and III meningiomas in mice [83]. Shortly after these initial studies, prostaglandin D2 synthase (PTGDS) was identified as a specific marker for meningioma precursor cells. By crossing transgenic mice expressing Cre under the *PTGDS* promoter to *Nf2*^flox/flox^ mice, biallelic *Nf2* inactivation in meningioma precursor cells was achieved, without the need for exogenous Cre delivery [84]. This resulted in the generation of meningothelial and fibroblastic meningiomas in the majority of animals. Taken together, these results provide proof for a fundamental role of merlin in meningioma induction and provide ideal models for further investigation into merlin signaling, especially incorporating insights from recent genetic studies based on next-generation sequencing.

## 6. Conclusions

Merlin/*NF2* loss is a key driver in the development of both syndromic and most sporadic meningiomas. Next-generation sequencing studies have provided a framework for an increasingly sophisticated categorization of meningiomas into two groups: non-*NF2* mutants and *NF2* mutants. It is intriguing, however, that the two types of *NF2*-inactivated meningiomas (type B and C) seem to have different underlying molecular mechanisms and dramatically different clinical outcomes. How the loss of merlin function leads to two very different biological dysregulations is yet to be investigated. These insights need to be further investigated through large-scale NGS studies and, more importantly, biochemical and molecular studies to reveal therapeutically relevant targets. 

## Figures and Tables

**Figure 1 cancers-11-01633-f001:**
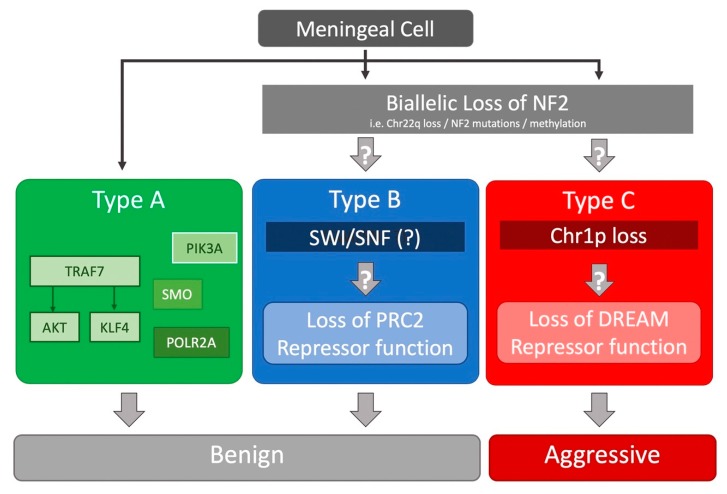
Summary of the mutational landscape of meningiomas based on several next-generation sequencing studies.

**Figure 2 cancers-11-01633-f002:**
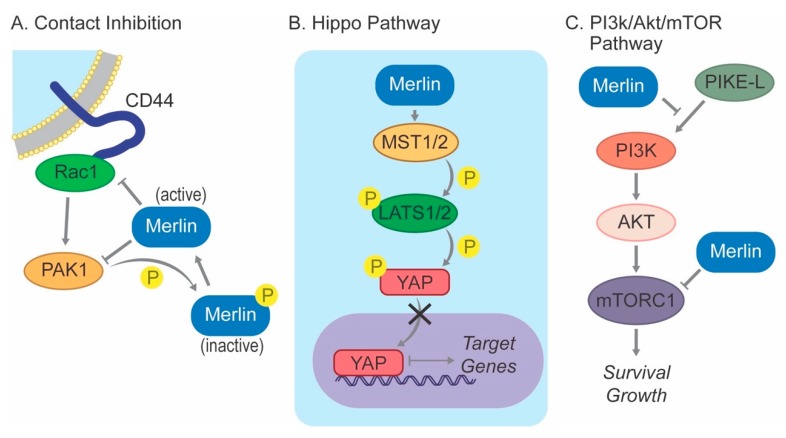
Merlin signaling. (**A**) Active, de-phosphorylated Merlin inhibits Rac1 and PAK1, key mediators of contact inhibition; (**B**) Merlin activates the Hippo pathway, leading to YAP phosphorylation and its sequestration from the nucleus; (**C**) Merlin is a negative regulator of PIKE-L and mTORC1.

**Table 1 cancers-11-01633-t001:** Manchester clinical diagnostic criteria for neurofibromatosis type 2 (NF2).

Diagnostic Criteria	Additional Findings Needed
Bilateral vestibular schwannomas	None.
Family history of NF2	Unilateral vestibular schwannoma, OR at least two of: meningioma, schwannoma, glioma, neurofibroma, cataract.
Unilateral vestibular schwannoma	At least two of: meningioma, schwannoma, glioma, neurofibroma, cataract.
Multiple meningiomas	Unilateral vestibular schwannoma, OR at least two of: schwannoma, glioma neurofibroma, cataract.

**Table 2 cancers-11-01633-t002:** Summary of next-generation sequencing (NGS) studies.

Study	Tumor Type (*n*)	Genetic Alterations	Key Findings
Clark *et al.* (2013) [36]	WHO I/II (243/57)	*NF2*/ch22q loss *TRAF7/KLF4* *TRAF7/AKT1* *SMO*	Mutually exclusive non-*NF2* driver mutations. *NF2* tumors are more aggressive. Non-*NF2* tumors are benign and localize to medial skull base.
Brastianos *et al.* (2013) [37]	WHO I/II/III (47/15/3)	*NF2*/ch22q loss *SMO* *AKT1*	As above.
Reuss *et al.* (2013) [38]	Secretory (30)	*TRAF7/KLF4*	All secretory meningiomas carried the *KLF4* K409Q mutation.
Clark *et al.* (2016) [39]	WHO I/II/III/? (552/214/7/2)	*POLR2A* *SMARCB1*	Identification of *POLR2A* driver mutation. *SMARCB1* and *NF2* mutations co-occur.
Agnihotri *et al.* (2017) [40]	Radiation-induced (31)	*NF2*/ch22q loss	*NF2* gene rearrangements common in radiation-induced tumors. Non-*NF2* driver mutations not observed.
Bi *et al.* (2017) [41]	WHO I/II/III (75/113/21)	*NF2*/ch22q loss Genomic instability	*NF2*/ch22q loss and genomic disruptions occur early in progression and remain consistent over time.
Harmanci *et al.* (2017) [42]	WHO I/II/III/? (548/211/7/9)	*NF2*/genomic instability *NF2/SMARCB1*	*NF2* is the sole driver mutation in atypical meningiomas and occurs in conjunction with genomic instability or *SMARCB1* mutations.
Patel *et al.* (2019) [43]	WHO I/II/III (119/33/5)	Loss of PRC2 or DREAM complex repression	Transcriptional signatures identified a sole subgroup with recurring tumors, characterized by DREAM target genes activation.

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
