# Peer review of "The Role of Merlin/NF2 Loss in Meningioma Biology"

_cancers, 2019, doi:10.3390/cancers11111633_

Round 1
Reviewer 1 Report
The authors present an interesting review on the role of Merlin on the meningioma biology. The review is complete for the biological and animal studies, but few paragraphs could be added by discussing:
how often meningiomas belong to the neurofibromatosis 2 spectrum and how often they do not. Cite in this regard the clinical criteria for the diagnosis of NF2, which are not reported, but could be useful for the reader add some genotype/phenotype of neurofibromatosis 2: do meningiomas always present in combination with vestibular schwannomas? are they more often single or multiple? is there any correlation with the other NF2 tumors, like the spinal tumors? The issue of mosaicism and somatic mutations, especially in sporadic cases with no family history, since negative NF2 genetic test results come also from patients with a clear clinical diagnosis of neurofibromatosis 2, and underline the importance of analysing for mutations also the surgical specimens, when available Cite also the peripheral schwannomatosis, even to exclude that they have ever shown meningiomas in their phenotype
The english language is fine. I have just find select instead of selected at page 2 line 55
Author Response
We thank the reviewer for these insightful comments and suggestions. We have added these points to an additional section on inherited disorders, covering both NF2 and Schwannomatosis. We have also added a table of the Manchester criteria for NF2.
Reviewer 2 Report
Dear authors, thank you very much for the nice overview describing the role of the NF2 gene/Merlin loss in meningiomas. Overall it is very well structured. Small notes that would surely improve the work are listed below.
Section 3"Insights From NGS Studies" reads very bumpy/complicated and the many information is a bit confusing. Here, an overview of the NGS studies as a table would certainly be helpful. Therein, the relevant studies, the tumor entity/group (e.g.secretory, radiation-induced, etc.), number of tumors, etc., as well as the relevant genes can be listed. The most important information can then be summarized in the section. Alternatively, the section should be better subdivided.
Author Response
We appreciate the reviewers suggestions. To this end, we have added a table (table 2) that summarizes these studies and hopefully helps add clarity to the section. We have also made some changes to the text to add clarity.